# OpenReview forum: "PerturbFormer: Adversarial Graph Transformers for Scalable and Resilient Representation Learning"
_ICLR.cc/2026/Conference — ICLR 2026 Conference Withdrawn Submission_

### Official Review · Reviewer_4Gtn · 2025-10-19

**Soundness:** 1
**Presentation:** 1
**Contribution:** 1
**Rating:** 2
**Confidence:** 4

**Summary:**

This paper proposes PerturbFormer, a transformer-based framework for graph learning that integrates multi-scale structural embeddings, contrastive pretraining, and adversarial propagation. By combining degree-normalized attention with generative perturbations, it achieves robust, efficient, and state-of-the-art performance across diverse graph tasks.

**Strengths:**

1. This paper proposes PerturbFormer, a new graph learning framework capable of handling multiple tasks, demonstrating good applicability across various domains.
2. The paper provides comprehensive experimental analyses, including evaluations on different tasks, ablation studies, efficiency assessments, and insightful visualizations.

**Weaknesses:**

1. The first major weakness of this paper lies in its writing quality. The manuscript is poorly written, and I suspect that the use of LLMs goes beyond typos and grammar (a review request has been submitted).
   (1) The Introduction fails to clearly convey the motivation and logical foundation of the proposed methods, while it lists up to six “limitations” the paper aims to address—many of which are exaggerated, such as the distillation limitation that is never discussed later.  (2) The Related Work section enumerates six points, but each is overly simplistic and resembles LLM-generated text. (3) In the Methodology section, the authors merely stack several modules without explaining their interconnections, and even use terms that sound machine-generated, such as “residual refinement” vs. “residual correction” or “feature extraction” vs. “feature synthesis,” which severely hampers readability. (4) Figure 1 provides little informative content and looks like an LLM generated it.

2. The PerturbFormer framework appears to be a combination of multiple existing modules and techniques (e.g., multi-resolution representation, contrastive alignment, adversarial propagation), yet their relationships are unclear. The approach PerturbFormer lacks novelty.

3. The experimental setup is confusing. For instance, PCQM4Mv2 is a graph regression dataset, but the paper incorrectly describes it as a node classification task. Table 3 lists datasets without any description or references.

4. The reported results are hard to believe: PerturbFormer achieves the best performance on all tasks and datasets, yet no standard deviation over multiple runs is provided, and no code for reproducibility.

5. Given that PerturbFormer integrates multiple modules, an analysis of its computational complexity is essential. However, no theoretical discussion is offered, and the results in Table 8 are unclear—comparing runtime across different methods and datasets is not meaningful.

Overall, the paper is logically inconsistent and poorly written, the proposed method lacks novelty, and there is concern that the use of LLMs in the writing and content generation exceeds acceptable limits, without any disclosure.

**Questions:**

Please respond to the Weaknesses part.

---

> ### Author Response · Authors · 2025-11-22
> **We sincerely hope to receive your support and encouragement!**
>
> **Dear Reviewer 4Gtn**, thank you for your careful reading and **constructive criticisms**. We appreciate the time you invested. Below we respond **point by point** to the weaknesses you raised, clarify **factual misunderstandings**, and describe **concrete revisions** and **additional analyses** we will add in the rebuttal and revision.
>
> # After carefully studying the comments, we realized that many points may include misunderstandings or inaccuracies, which may have contributed to a lower evaluation of our submission.
> # We sincerely hope to receive your support and encouragement!
> # 1.  “Writing quality is poor; possible excessive LLM use”
>
> **Response.** We will substantially improve **exposition**, **notation**, and **figure design** in the revised manuscript. **Concretely:**
>
> We will move the **compact overview diagram** (currently in the Appendix) into the **main text**, redesign **Figure 1** to clearly show **module interactions** (generator ↔ propagation ↔ attention ↔ fusion), and add **Algorithm 1** inline for immediate clarity.
>
> We will rework **Section 1** so the **six limitations** are grouped into **three logically coherent categories** (**propagation/adaptation limits**, **structural/statistical brittleness**, **practical/scalability constraints**) and explicitly map each category to the corresponding **design choice**.
>
> Regarding **LLM usage**: the **scientific design**, **experiments**, **proofs**, and **code** are authored by the paper’s authors. Any **minor language polishing** performed for **grammar/readability** was **editorial only** and did not alter **technical content**. We will add an **explicit statement** in the **acknowledgments/author notes** disclosing any **editorial assistance** used for **language polishing**.
> # 2. “The distillation limitation is never discussed later”
> **Response (correction and plan).** This is a **misunderstanding**. The **distillation/condensation limitation** appears in **Section 1** to motivate the need for **condensed, multi-scale representations**. We discuss **concrete architectural and training remedies** in **Section 3** (**multi-hop fusion** and **condensed representations**) and in **Appendix A.1** (**implementation details**. Nevertheless, we acknowledge the current wording could be clearer. In the revision we will **expand the short paragraph** linking **distillation limitations** to our **multi-scale condensation design** and **add a short ablation** that contrasts a **distilled/condensed variant** to the **full model** to **quantify the trade-off**.
> # 3. “Method is just stacked modules; relationships unclear; lacks novelty”
>
> **Response (firm rebuttal and evidence).** This is **not correct**. **PerturbFormer** introduces **tightly coupled mechanisms** whose **joint behavior** is **non-trivial** and **empirically non-additive**:
>
> **Adaptive confidence-gated residual propagation** (Eq. below) , a **learned per-node gate** $$c_i$$ that controls how much **neighborhood information** each node accepts:
>
> $$
> R(t+1) = (I - \text{diag}(c)) R(0) + \text{diag}(c) \,\tilde{A}\, R(t).
> $$
>
> We provide a **contraction / fixed-point analysis** (**Appendix B, Theorem 1**) proving **convergence under mild spectral conditions**; the theorem connects **learned confidences** to **provable stability** and **error bounds**.
>
> **Propagation-aware adversarial generator** , a **generator conditioned on current embeddings** that synthesizes **structured topological perturbations** (not generic random noise). The generator’s **objective and conditioning** make the adversary directly **target the propagation operator**; the resulting **adversarial examples** are therefore **propagation-relevant** and produce **robustness** that **simple stacking cannot achieve**.
>
> **Heterophily-aware attention and fusion** , an **attention bias** tailored to **heterophilous neighborhoods** and a **principled fusion** of **diffusion and transformer streams** that preserves **expressivity** while preventing **harmful over-smoothing**.
>
> Crucially, **ablation studies** (**Section 4.3, Table 4**) show that **removing the GAN or the adaptive propagation** causes **substantial performance drops** (e.g., **Full model 86.40 → w/o GAN 84.25 / w/o adaptive propagation 84.91 on Proteins**), demonstrating **non-additive synergy** rather than **marginal stacking**. We will make this point **more explicit** in the **revision** and place the **ablation table** nearer the **main text** for clarity.

---

> ### Author Response · Authors · 2025-11-22
> **We sincerely hope to receive your support and encouragement!**
>
> # 4. “PCQM4Mv2 incorrectly described as node classification; Table 3 lacks dataset descriptions”
> **Response (correction).** This assertion is **incorrect**: **PCQM4Mv2** is correctly used and described as a **graph-level regression benchmark** in our submission (see **Section 4.1** and **Appendix dataset table**). We apologize if the current placement of **dataset descriptions** caused confusion. To eliminate ambiguity we will **relocate the dataset description table** (with **citations** and **task types**) into the **main experimental section**, and **add a one-line label** next to each **dataset in Table 3** clarifying the **task type** (**node/graph/edge** and **regression/classification**).
> # 5. “Reported results look suspicious: no standard deviations, no code”
>
> We appreciate this **important point**. All reported results are computed on **held-out test sets** with **validated hyperparameters**; however, we agree that **statistical reporting** and **reproducibility** can be improved. We will:
>
> - **Run each main experiment with 5 random seeds** (same **training/validation split protocol**) and report **mean ± standard deviation** for all tables in the **revised paper**.
> - **Release code**, **training scripts**, and **exact hyperparameter files** on a **public repository** at or before the **camera-ready stage**.
>
> Currently, we have already provided the **main code** in the **supplementary materials**.
>
> # 6. “Computational complexity not analyzed; Table 8 unclear”
>
> **Response (clarify and add formal analysis).** We will add a **concrete complexity analysis** in the **Appendix** and **summarize** it in the **main text**. Below is a **concise complexity statement** we will include:
>
> Let **n** be the **number of nodes**, **m** the **number of edges**, **d** the **hidden dimension**, **L** the **number of transformer/diffusion layers**, and **h** the **number of attention heads**. Then:
>
> - **Multi-head attention** (sparse/degree-normalized) per layer:
>   $$O(md + nd^2 / h)$$
>   We implement **attention using edge lists and degree-normalization** so cost scales with **m** not **n²**.
>
> - **Multi-hop diffusion / propagation** per iteration:
>   $$O(md)$$ using **sparse matrix–vector operations**.
>
> - **Generator / discriminator** (topology perturbation) per update:
>   $$O(m' d)$$ where **m'** is the **number of edges touched by generator** (a small fraction of **m**).
>
> - **Total per-epoch approximate cost**:
>   $$O(L \cdot (md + nd^2))$$ (constants depend on **heads** and **fusion operations**).
>
> We will **add these formulas**, provide **actual runtime-per-epoch** and **memory usage normalized by m or n** (instead of raw seconds), and **expand Table 8** to show **normalized throughput** (**samples/sec** and **FLOPs estimate**) for **fairer cross-method comparison**.
> # 7.“Figure 1 uninformative; Table 3 lacks references”
>
> **Response (fix).** We will:
>
> - **Replace Figure 1** with a **clearer overview** that labels **dataflow**, **per-module inputs/outputs**, and **training objectives** (**contrastive pretraining**, **GAN**, **propagation loss**). The updated figure will be **visually simplified** and **annotated**.
> - **Expand Table 3** with **dataset citations** and **short descriptions**.
> Table 3 **actually refers to the methods compared above**.
> # 8. “No theoretical discussion of computational complexity / algorithmic guarantees”
> **Response (add and point to proofs).** We already provide **formal convergence results** for the **adaptive operator** (**Appendix B**) and will **add the complexity analysis** described above. We will also **add a short paragraph** discussing **practical mitigations** (**sparse implementation**, **neighbor sampling**, **mixed precision**) that preserve **scalability**.

---

> ### Author Response · Authors · 2025-11-22
> **We sincerely hope to receive your support and encouragement!**
>
> # 9. Summary
> We thank **Reviewer 4Gtn** for the **detailed critique**. We **respectfully disagree** with several **factual claims** (**PCQM4Mv2** is correctly treated as **graph regression**; the paper contains **formal contraction proofs** for the **adaptive propagation operator** see **Appendix B**) and **accept constructive points** about **exposition**, **reproducibility**, and **complexity reporting**. In the revision we will:
>
> - **Substantially rewrite** the **Introduction** and **Related Work** for **clarity** and **remove ambiguous wording**.
> - **Move the overview diagram** and **Algorithm 1** into the **main text**.
> - **Add mean ± std** over **multiple seeds** and **release code**.
> - **Add GAN stability diagnostics** and **embedding-consistency plots** in the **supplement**.
> - **Include a formal complexity analysis** and **normalized runtime metrics** so **comparisons in Table 8** are **meaningful**.
>
> The paper’s **scientific contributions** **propagation-aware adversarial generation**, **confidence-gated residual propagation with provable stability**, and **heterophily-aware attention/fusion** remain **novel** and supported by **ablation** and **theoretical analysis** (**Section 3** and **Appendix B**). We will make all of these **clarifications** and **additions** in the **final revision**.

---

> ### Author Response · Authors · 2025-11-22
> **proof**
>
> # Proposition 1: Computational complexity of a self-attention layer
>
> **Statement.**
> A single **dense (full) self-attention layer** applied to **N tokens (nodes)** with **feature dimension d** has dominant time complexity **O(N² · d)**. If attention is computed only over **graph edges (sparse attention)** with **E edges**, the dominant cost becomes **O(E · d)**.
>
> **Proof (steps).**
> A standard self-attention layer performs three dominant operations:
>
> ### **Linear projections:** Input **X ∈ ℝ^{N×d}** is projected to **queries Q**, **keys K**, and **values V**, each of dimension **d'** per token. Computing these three linear maps costs **O(N · d · d')**. For typical settings **d' = O(d)**, we may treat this step as **O(N · d²)** or **O(N · d)** if **d** is treated constant.
>
> ### **Attention score computation:** Dense attention computes the matrix of inner products **QKᵀ**, which involves **N × N** inner products, each of cost **O(d')**. So this step costs **O(N² · d')**.
>
> ### **Weighted sum:** Multiplying the **N×N attention matrix** by **V (N×d')** to obtain outputs again costs **O(N² · d')**.
>
> The dominant term is **O(N² · d')**. With **d' = O(d)**, this is **O(N² · d)**. If attention is restricted to a **sparse neighborhood** (only compute scores for edge pairs), the number of pairs reduces from **N²** to **O(E)**, and the two quadratic steps become **O(E · d')**, hence **O(E · d)**. ∎
>
> ---
>
> # Theorem 2: Lipschitz upper bound for pointwise adversarial loss
>
> **Statement.**
> Assume for fixed model parameters **θ** the per-example loss **L(x) := ℓ(f_θ(x), y)** is **L-Lipschitz** in input **x** (so |L(x) − L(x')| ≤ L · ||x − x'|| for all x,x'). For adversarial perturbations **δ** with ||δ|| ≤ ε, the following holds for any **x** and **θ**:
> $$
> \sup_{\|\delta\| \le \varepsilon} \ell(f_\theta(x + \delta), y) \le \ell(f_\theta(x), y) + L \cdot \varepsilon.
> $$
>
> **Proof.**
> By the **L-Lipschitz property**, for any **δ** with ||δ|| ≤ ε we have:
> $$
> \ell(f_\theta(x + \delta), y) - \ell(f_\theta(x), y) \le L \cdot \|(x + \delta) - x\| = L \cdot \|\delta\| \le L \cdot \varepsilon.
> $$
> Rearranging gives
> $$
> \ell(f_\theta(x + \delta), y) \le \ell(f_\theta(x), y) + L \cdot \varepsilon
> $$
> for every such **δ**. Taking the supremum over all admissible **δ** yields
> $$
> \sup_{\|\delta\| \le \varepsilon} \ell(f_\theta(x + \delta), y) \le \ell(f_\theta(x), y) + L \cdot \varepsilon.
> $$
> This completes the proof.
>
> ---
>
>
> **Interpretation.**
> If the **loss is Lipschitz** in the input, the **adversarial (worst-case) pointwise loss** is at most the **clean loss plus L·ε**. Thus **adversarial training**, which seeks to minimize the supremum on the left, controls **worst-case degradation** in terms of **L** and **ε**.

---

> > ### Author Response · Authors · 2025-11-22
> > **We sincerely hope to receive your support and encouragement!**
> >
> > **Proposition 3: Multi-scale (k-hop) information can strictly increase discriminative power**
> >
> > **Statement (constructive).**
> > There exist **graphs** and **node feature assignments** for which two nodes are **indistinguishable by any 1-hop local GNN** but become **distinguishable if the model uses 2-hop (or larger-scale) structural information**.
> >
> > **Constructive proof (explicit small example).**
> > Consider the following small graph and features:
> >
> > - **Nodes:** {u, v, a, b}
> > - **Edges:** (u, a), (u, b), (v, a). So **u** is connected to **a** and **b**; **v** is connected only to **a**.
> > - **Node features:** xᵤ = 0 (zero vector), xᵥ = 0, xₐ = e₁, x_b = e₂ (distinct nonzero vectors).
> >
> > Compute **1-hop aggregated features** using simple sum-aggregation:
> >
> > - For **u:** 1-hop sum = xₐ + x_b = e₁ + e₂
> > - For **v:** 1-hop sum = xₐ = e₁
> >
> > If we want **u** and **v** to be indistinguishable at 1-hop, we can modify features so their 1-hop sums coincide: set x_b = 0 (zero). Then both 1-hop sums equal e₁. Thus any model relying only on node features and unlabeled 1-hop aggregation (i.e., any 1-hop GNN whose output for a node is a function of that node's feature and the multiset of its 1-hop neighbor features) will produce identical representations for **u** and **v**.
> >
> > Now examine **2-hop neighborhoods**:
> >
> > - **u's 2-hop neighbors** include neighbors of **a** and **b**; in our graph, **u's 2-hop set** includes nodes reachable in two steps (which can include **v** through **a**), while **v's 2-hop neighborhood** differs in structure and/or features in general. In particular, with x_b set to 0 and xᵥ = 0, the pattern of distances and multiplicities in the 2-hop neighborhood differs between **u** and **v**. A function that inspects the multiset of 2-hop features (or counts paths of length 2) can distinguish **u** and **v**.
> >
> > Therefore there is an explicit graph and feature assignment where 1-hop information is identical but 2-hop information differs, showing **multi-scale input strictly increases representational power**.
> >
> > **Conclusion.**
> > This constructive example proves that including **k-hop (k ≥ 2) information** enlarges the class of node equivalence classes a model can distinguish, i.e., **multi-scale embeddings strictly increase discriminative capability in general**.
> >
> > ---
> >
> > **Lemma 4: Contrastive/aligning regularization reduces effective hypothesis complexity (sketch via Rademacher complexity)**
> >
> > **Statement (informal but standard).**
> > Adding a **contrastive or alignment regularizer** that enforces **pairwise similarity constraints** on learned representations effectively **reduces the hypothesis space** of allowed functions. Under standard **generalization bounds based on Rademacher complexity**, this reduction yields a **tighter generalization bound**.
> >
> > **Sketch of argument.**
> > Let **F** be the original function class (mappings from inputs to scores/representations). The **Rademacher complexity Rₙ(F)** measures the richness of **F** on **n samples** and appears in generalization bounds of the form:
> > $$
> > \text{expected risk} \le \text{empirical risk} + O(R_n(F)) + \text{small terms}.
> > $$
> >
> > Introducing a **contrastive penalty** (for example, a term that requires representations of positive pairs to be within margin τ and negatives to be separated) can be seen as restricting **F** to the subset **F_λ ⊂ F** of functions that satisfy the contrastive constraints up to strength **λ** (**λ** parameterizes how strongly the constraint is enforced). Intuitively, **F_λ** has fewer degrees of freedom than **F**, so its **Rademacher complexity Rₙ(F_λ) ≤ Rₙ(F)**. If the contrastive constraint is nontrivial, one can often show
> > $$
> > R_n(F_\lambda) \le \alpha(\lambda) \cdot R_n(F)
> > $$
> > for some **α(λ) < 1** that decreases as **λ** increases (this follows from standard arguments that constraints or norm bounds shrink complexity).
> >
> > Therefore, with the **contrastive regularizer active**, the generalization bound becomes:
> > $$
> > \text{expected risk} \le \text{empirical supervised loss} + \text{contrastive regularization term} + O(R_n(F_\lambda)),
> > $$
> > and since **Rₙ(F_λ) ≤ Rₙ(F)** this yields a **potentially tighter bound** on expected risk. Thus **contrastive pretraining or joint contrastive regularization** acts as a form of **statistical regularization** that reduces **overfitting risk**.

---

> ### Author Response · Authors · 2025-11-22
> **We sincerely hope to receive your support and encouragement!**
>
> #  Research context and open gaps
> Graph learning today broadly follows two complementary paradigms:
>
> - **Local message-passing GNNs** (e.g., GCN, GraphSAGE, GAT) are efficient and effective on many benchmarks but suffer from **limited receptive field** and **difficulty modeling long-range or multi-resolution structure**. They also can **oversmooth** or fail on **structural tasks** where **higher-order substructures** are critical.
>
> - **Transformer-style graph models** (e.g., Graphormer and variants) leverage **global attention** and **positional encodings** to capture **long-range interactions**, but **dense attention is O(N²)** in time/memory and can be **biased by node degree**. Naive adoption of transformer machinery also leaves open how to inject **topology-aware robustness** and **multi-scale structure** in a principled way.
>
> In parallel, **self-supervised/contrastive pretraining** and **adversarial/generative perturbation methods** have proven effective in vision and language for improving **robustness** and **generalization**, yet there is no widely accepted unified framework that:
> - combines **multi-scale structural representation** with **transformer-style global modeling**,
> - trains with **generative adversarial perturbations** tailored to **graph structure**, and
> - scales and applies across **node/edge/graph prediction tasks** while providing **practical reproducibility** and **complexity accounts**.
>
> These gaps motivate **PerturbFormer**.
>
> ---
>
> #  Motivation
> Our central motivation is to build a single, practical architecture that simultaneously:
>
> - **extends representation power** beyond **single-hop aggregation** via **explicit multi-scale structural embedding**,
> - **retains the expressive benefits of Transformer attention** while correcting **degree-related bias** and controlling **complexity**, and
> - **improves worst-case and distributional robustness** by incorporating **learned generative perturbations** during **propagation** as a **training-time regularizer**.
>
> Practically, this addresses three commonly occurring shortcomings in graph learning: **limited multi-scale discrimination**, **brittle behavior under structured perturbations**, and **inconsistent scaling of Transformer-based approaches on large graphs**.
>
> ---
>
> #  Key innovations (technical points)
>
> - **Multi-scale structural embedding + degree-normalized Transformer attention**
>   We combine **explicit multi-resolution structural descriptors** (k-hop, spectral-ish summaries) with a **Transformer attention kernel modified for degree normalization**. This pairing preserves **global relational modeling** while controlling **degree-induced attention bias**.
>
> - **Generative perturbation module integrated into propagation (generative adversarial propagation)**
>   Unlike standard adversarial training or random dropout, we train a **small generator** that proposes **plausible structure-aware perturbations**. A **discriminative/robustness objective** forces the encoder to learn **representations stable under realistic structural perturbations**.
>
> - **Confidence-weighted residual correction during propagation**
>   To avoid **blind aggregation of noisy or adversarial signals**, we introduce a **residual correction mechanism** that weights **propagated updates** by **learned per-node confidence scores**. This yields a **principled way to attenuate harmful perturbation signals** while preserving **useful multi-scale information**.

---

> ### Author Response · Authors · 2025-11-22
> **We sincerely hope to receive your support and encouragement!**
>
> - **Engineering practices for scalability and reproducibility**
>   We combine **sparse/edge-restricted attention** when appropriate, **mixed-precision training**, and **checkpointing strategies**, and we provide a **standardized evaluation protocol** across **node/edge/graph tasks** including **explicit measurement protocols for runtime and memory**.
>
> ---
>
> # Concrete contributions
>
> - A **unified architecture (PerturbFormer)** that combines **multi-scale structural embeddings**, **degree-normalized transformer attention**, **generative perturbations**, and **confidence-weighted residual correction**; applicable to **node, edge, and graph-level tasks**.
>
> - **Theoretical clarifications and practical complexity analysis**
>   Dense attention **O(N²d)** vs edge-limited **O(E d)**; **Lipschitz-based upper bound** that justifies **adversarial training** for **worst-case risk control**; a **constructive argument** that **multi-scale features strictly increase discriminative power in graphs**. These are provided in the **appendix**.
>
> - A **comprehensive empirical evaluation** across **heterogeneous graph tasks**, accompanied by **ablations** that quantify the **marginal contribution of each module**. We commit to **reporting mean ± standard deviation across multiple seeds** and to **providing the reproduction package**.
>
> - **Practical techniques and measurements** to make **transformer-style graph modeling usable at scale** (sparse attention, cost-tradeoff discussions, and unified timing measurement protocol).
>
> ---
>
> #  Why this work fits ICLR 2026 themes
>
> - **Representation learning and self-supervised learning:** Our **contrastive pretraining and alignment design** contribute to **representation learning theory and practice**.
> - **Learning on graphs and geometries:** The architecture specifically targets **graph-structured data** and **multi-scale topology**.
> - **Generative models and robust learning:** The **generative perturbation module** connects **generative modeling** to **robust representation learning**.
> - **Learning theory and optimization:** We provide **complexity analysis** and **risk-control arguments** for **adversarial objectives**.
> #  We firmly believe that with your suggestions, our paper will be further improved, and we sincerely hope your score improvement.

---

> ### Author Response · Authors · 2025-11-27
> **We sincerely hope to receive your support and encouragement!！**
>
> # We have addressed the reviewer's concerns and improved our approach. Thank you very much for all the reviewers' suggestions. The latest version has been uploaded and we hope to receive the support and encouragement of all the reviewers, and we sincerely hope your score improvement！

---

### Official Review · Reviewer_gH8g · 2025-10-26

**Soundness:** 4
**Presentation:** 3
**Contribution:** 4
**Rating:** 8
**Confidence:** 4

**Summary:**

This paper addresses key challenges in graph-based semi-supervised learning, including heterophily, robustness to structural noise, and scalability. It proposes PerturbFormer, an integrated framework that synthesizes multi-scale feature augmentation, contrastive self-supervised pretraining, generative adversarial networks, and optimized transformer architectures. The methodology involves enhancing node representations through multi-hop structural embeddings, adversarial contrastive pretraining, and a Graphormer with degree-normalized attention. Core contributions include a novel adversarial propagation mechanism for dynamic perturbation synthesis and confidence-weighted residual refinement. Experimental evaluation on benchmarks demonstrates improved classification accuracy and robustness compared to state-of-the-art baselines.

**Strengths:**

1.	The framework is novel as it combines feature synthesis, contrastive alignment, and GAN-based perturbations to enhance robustness.
2.	Convergence guarantees for residual propagation are provided.
3.	The paper is easy to follow.

**Weaknesses:**

1.	Claims of outperforming prior work in low-homophily settings are not rigorously justified with comparative analysis; evidence is limited to benchmark results in Table 2 without discussing fundamental gaps.
2.	The GAN-based perturbation lacks analysis of critical issues like model training stability. There is no experimental curves or metrics on adversarial optimization progress.

**Questions:**

See weaknesses.

---

> ### Author Response · Authors · 2025-11-22
> **Thank you very much for your support and assistance to us! You are our shining light!**
>
> # Thank you very much for your support and assistance to us! You are our shining light!
> Dear Reviewer gH8g, thank you for the constructive reading and for recognizing the paper’s strengths. We appreciate the positive assessment and the suggestions. Below we respond to the two main concerns you raised.
> # We greatly appreciate any support and assistance you have provided to our work!
> # 1.“Claims of outperforming prior work in low-homophily settings are not rigorously justified.”
>
> **Response (clarify and evidence)**
> We agree that claims should be supported by both **empirical and conceptual evidence**, and they are. Concretely:
>
> **Empirical evidence**
> **Table 2** (Node classification across five benchmarks) shows that **PerturbFormer yields substantially larger gains on biological interaction networks (Proteins)**, a dataset we explicitly identify as **challenging due to heterophily and label sparsity**. For example, on **Proteins our model achieves 86.40% vs. Graphormer’s 72.17%** (Table 2, Section 4.2.2). These are not isolated wins: the **ablation suite (Section 4.3, Table 4)** demonstrates that **heterophily-aware attention and confidence-gated propagation are the subcomponents responsible for these gains** (removing them reduces performance on Proteins substantially).
>
> **Theoretical and mechanistic analysis**
> Beyond raw numbers, we explicitly analyze **why standard assumptions break down in heterophilous graphs and how our design addresses this**. **Appendix B and Section 3** examine **spectral conditions** (we show via **SBM experiments that heterophilous graphs often violate ∥A∥₂ ≤ 1, Fig. 8**) and derive a **snapshot-wise contraction condition**:
>
> **supₜ maxᵢ cᵢ(τ) ⋅ ∥Ãτ∥₂ < 1**
>
> which links the **learned per-node confidences cᵢ and normalized adjacency Ã** to **provable iterative stability** (Appendix B). The **heterophily-aware attention bias reduces harmful cross-class mixing**, while the **confidence-gated residual stops low-confidence nodes from amplifying noisy signals**. These two mechanisms are the **conceptual explanation for the empirical gains on low-homophily graphs** (see Section 3.4–3.6 and Appendix B).
>
> **Summary**
> Our claims are supported by **benchmark results that emphasize low-homophily improvements (Table 2)**, **ablation evidence that isolates the heterophily components (Table 4)**, and **targeted spectral/SBM analysis that explains why the method addresses fundamental gaps (Section 3 and Appendix B)**. We will **make these links more explicit and bring the spectral/SBM discussion earlier in the revised manuscript to further clarify this connection**.

---

> ### Author Response · Authors · 2025-11-22
> **Thank you very much for your support and assistance to us! You are our shining light!**
>
> # 2. “GAN-based perturbation lacks analysis of training stability, no optimization curves or metrics on adversarial progress.”
>
> **Response (concrete pointers and metrics already reported)**
> We thank the reviewer for highlighting this important point. The submission already contains multiple analyses demonstrating **stable and meaningful adversarial training**:
>
> **Algorithm and objective**
> Our adversarial module uses a **Wasserstein objective with gradient penalty (WGAN-GP)**. The paper details the **generator/discriminator architecture and optimization objective** (Section 3.6). **WGAN-GP is chosen specifically for its improved training stability in practice**.
>
> **Quantitative stability metrics**
> Rather than only showing optimization traces, we report **representational and parameter stability metrics** that quantify the effect and stability of adversarial training:
> - **Embedding-consistency**: We measure **embedding distortion** and report a **40.7% reduction compared to the ablated variant** (Section 4.4, Figure 2).
> - **Parameter stability**: We track **parameter stability over snapshots in temporal experiments** (Table 6).
> These metrics directly reflect whether the **adversarial loop produces stable, robust embeddings** and whether **optimization behaves well across training**.
>
> **Controlled perturbation analysis**
> **Appendix C (Section C.1)** provides an analysis of **edges modified by the generator at perturbation levels δ ∈ {5%, 10%, 15%}** and reports **topological statistics (degree shifts, clustering changes)**. This demonstrates that the **generator produces interpretable, structured perturbations rather than arbitrary noise**. Paired with the **embedding-consistency results**, this indicates **adversarial training is both meaningful and controlled**.
>
> **Monitoring practice**
> In our experiments we monitor **generator/discriminator WGAN-GP losses**, **gradient-penalty terms**, and the **embedding-distortion metric**. These traces were **stable across checkpoints for all reported runs**. To address the reviewer’s suggestion, we will **add representative optimization curves (generator loss, discriminator loss, gradient penalty, and embedding distortion over epochs) to the supplementary material** so readers can directly inspect **adversarial optimization dynamics**.
>
> **Summary**
> We already use a **stability-oriented GAN objective (WGAN-GP)**, report **embedding and parameter-stability metrics** that empirically demonstrate **stable adversarial training**, and analyze the **structure of generated perturbations**. We will **include explicit optimization traces and a short discussion of hyperparameters (e.g., GP weight, discriminator steps per generator step) in the appendix** to make the stability story fully transparent.
>
> # 3. Conclusion
>
> We will **make the heterophily → method → empirical link clearer in the main text** by bringing **key spectral analysis and SBM results into Sections 2 and 3**, **add representative GAN optimization curves and explicit hyperparameter choices for adversarial training to the supplementary material**, and **emphasize the ablation evidence that isolates heterophily-aware and adversarial components**. The manuscript already contains the **formal contraction result (Appendix B)**, **perturbation analyses (Appendix C)**, **embedding-consistency metrics**, and the **dataset results (Table 2, Table 4, Table 6)**, all accessible in the submission file.
> **Thank you again for the careful review and for suggestions that will strengthen the paper.**

---

> > ### Author Response · Authors · 2025-11-22
> > **proof**
> >
> > # A. Propagation adaptivity reduces error under heterophily, theorem and proof
> >
> > ### Setting and Notation (Concise)
> >
> > **Graph normalized adjacency (or diffusion) matrix**:
> > **Ã ∈ ℝⁿ×ⁿ**, with **∥Ã∥₂** denoting its spectral norm.
> >
> > **Learned per-node confidence vector**:
> > **c ∈ (0,1)ⁿ**; denote **D = diag(c)**.
> >
> > **Initial (noisy) per-node signals** (labels or logits) used to start residual refinement:
> > **R(0) ∈ ℝⁿ×ᵈ**. True target signals are **Y* ∈ ℝⁿ×ᵈ**. We analyze per-column and use vector notation for simplicity.
> >
> > ---
> >
> > ### Iterative Update (Adaptive Residual Operator)
> >
> > **R(t+1) = (I − D)R(0) + D Ã R(t)**
> >
> > Define the mapping:
> > **T(R) = (I − D)R(0) + D Ã R**
> >
> > ---
> >
> > ### Theorem 1 (Existence, Fixed Point, and Error Bound)
> >
> > Assume **∥D Ã∥₂ < 1**. Then:
> >
> > 1. **T is a contraction** and the iteration converges to the unique fixed point:
> >    **R⋆ = (I − D Ã)⁻¹ (I − D)R(0)**
> >
> > 2. The steady-state estimation error relative to the true signal **Y***,
> >    **E := R⋆ − Y***, satisfies the bound:
> >    ∥E∥₂ ≤ ∥I − D∥₂ / (1 − ∥D Ã∥₂) ⋅ ∥R(0) − Y*∥₂
> >
> > ---
> >
> > ### Proof Sketch
> >
> > - Since **∥D Ã∥₂ < 1**, the linear operator **R ↦ D Ã R** has spectral radius < 1, so **I − D Ã** is invertible and **T** is a contraction (Lipschitz constant **∥D Ã∥₂**).
> > - The iteration converges to the unique fixed point solving:
> >   **R⋆ = (I − D)R(0) + D Ã R⋆**, i.e.,
> >   **R⋆ = (I − D Ã)⁻¹ (I − D)R(0)**.
> > - For the error bound:
> >   E = R⋆ − Y* = (I − D Ã)⁻¹ [(I − D)R(0) − (I − D Ã)Y*] = (I − D Ã)⁻¹ (I − D)(R(0) − Y*)
> > - Apply norm bounds and Neumann series:
> >   **∥(I − D Ã)⁻¹∥₂ ≤ 1 / (1 − ∥D Ã∥₂)**, giving the stated inequality.
> >
> > ---
> >
> > ### Interpretation and Corollary (Why Adaptivity Helps Under Heterophily)
> >
> > If confidences **cᵢ** are smaller for nodes whose local neighborhoods are unreliable (e.g., heterophilous neighbors or suspected noisy nodes), then **∥I − D∥₂** decreases and **∥D Ã∥₂** also reduces (because **D** down-weights rows/columns associated with unreliable nodes), tightening the multiplicative constant **∥I − D∥₂ / (1 − ∥D Ã∥₂)**.
> >
> > Concretely, compared to uniform gating **D = c₀ I**, adaptively setting small **cᵢ** on noisy nodes yields strictly smaller **∥D Ã∥₂** and often smaller numerator **∥I − D∥₂**, hence a smaller upper bound on **∥E∥₂**.
> >
> > Extreme case: if **cᵢ = 0** for all truly noisy nodes, then those nodes do not accept neighborhood propagation at all, reducing local error amplification and preventing propagation of corrupted signals across the graph.
> >
> > This provides a **formal explanation why per-node learned gating is provably beneficial compared to uniform propagation in settings where certain nodes have unreliable neighbors**, a situation typical in **low-homophily graphs**.

---

> ### Author Response · Authors · 2025-11-22
> **GAN perturbation training: stability statement, sketch of proof, and recommended diagnostics**
>
> # B. GAN Perturbation Training: Stability Statement, Proof Sketch, and Recommended Diagnostics
>
> Below we provide a **theoretical justification** and **practical monitoring plan** for the **WGAN-GP style objective** used for topology perturbation, plus a short **proof sketch** showing that the **gradient penalty enforces approximate 1-Lipschitzness**, which yields **stable discriminator gradients** and **meaningful generator updates**.
>
> ---
>
> #### Setup (Objective)
>
> Let **Pᵣ** be the empirical distribution of **clean graphs or adjacency structures**, and let **Gϕ** induce a distribution **Pϕ** of **perturbed topologies** (edge flip patterns or reweighted adjacency). The discriminator **fθ** is trained with the **WGAN-GP objective**:
>
> **minϕ maxθ  Eₓ∼Pᵣ[fθ(x)] − Eₓ̂∼Pϕ[fθ(x̂)] − λ Eₓ̃∼P[(∥∇ₓ̃ fθ(x̃)∥₂ − 1)²]**
>
> where **P** is the distribution for computing the gradient penalty (interpolations), and **λ > 0**.
>
> ---
>
> #### Proposition 2 (WGAN-GP Yields Stable Discriminator Gradients)
>
> Assume **fθ** has sufficient capacity and **λ** is chosen so that the gradient-penalty term is enforced during training. Then, during discriminator updates, **fθ** is approximately **1-Lipschitz** in regions interpolating between **Pᵣ** and **Pϕ**. Consequently, the discriminator gradients with respect to generator parameters **∇ϕ L_G** are **well-behaved (bounded)**, which prevents exploding gradients for the generator and produces **meaningful, stable update directions**.
>
> ---
>
> #### Sketch of Argument
>
> - **WGAN duality**: With a **1-Lipschitz discriminator class**, the inner max recovers a lower bound to the **Wasserstein-1 distance W(Pᵣ, Pϕ)**. Gradient directions from discriminator to generator correspond to **transport directions between distributions**.
> - **Gradient penalty term** penalizes deviations **∥∇f∥₂ − 1** on interpolations, explicitly encouraging Lipschitzness in sample paths between real and generated distributions (Gulrajani et al., 2017). Under mild regularity, this keeps **∥∇f∥ bounded near 1**.
> - **Bounded critic gradients imply bounded ∇ϕ L_G** (via chain rule through generator), preventing unstable large updates and enabling **stable alternating optimization** (in practice, combined with learning rate control and discriminator steps per generator step).
>
> While full formal convergence to a global Nash equilibrium requires strong assumptions, the combination of **WGAN objective plus gradient penalty** is a widely used practical choice that yields **stable, informative gradients for generator learning** — exactly the property we need when generating **structured perturbations meaningful to the propagation operator**.
>
> ---
>
> ### Practical Diagnostics and Extra Plots / Metrics to Add
>
> To address the reviewer’s request for **training-stability analysis**, include the following in the supplement:
>
> #### **WGAN losses over epochs**
>    Plot **discriminator (critic) loss**:
>    **Eₓ∼Pᵣ[fθ(x)] − Eₓ̂∼Pϕ[fθ(x̂)]**
>    and **generator loss**:
>    **−Eₓ̂∼Pϕ[fθ(x̂)]**.
>    Show they trend and settle (not oscillate wildly).
>
> #### **Gradient penalty term**
>    Plot **Eₓ̃[(∥∇ₓ̃ fθ(x̃)∥₂ − 1)²]** over training. Stable training keeps this term small (close to zero) after initial transients.
>
> ### **Embedding-consistency (key empirical diagnostic)**
>    Define embedding distortion under perturbation level δ as:
>    **Dist(δ) = (1/n) Σᵢ ∥zᵢ − zᵢ^δ∥₂² / Var({zⱼ})**
>    where **zᵢ** is the node embedding under the clean graph and **zᵢ^δ** under generator-induced perturbation.
>    Plot **Dist(δ)** over epochs for a fixed δ (e.g., 10%), showing the generator produces **structured perturbations** while the encoder+propagation maintain **bounded distortion**.
>
> #### **Parameter stability / performance tracking**
>    Plot **validation ROC-AUC vs. epoch** together with **embedding distortion** to show downstream performance remains stable or improves as adversarial training progresses.
>
> #### **Topological statistics of generated perturbations**
>    Report summary stats: fraction of edges flipped, per-node degree change distribution, clustering coefficient change, and class-wise edge-crossing ratio. Consistent, interpretable stats indicate the generator is not producing random noise.
>
> #### **Ablation: discriminator steps and GP weight**
>    Short grid: discriminator steps per generator step ∈ {1, 5} and GP weight λ ∈ {1, 10}. Show relative stability to justify chosen hyperparameters.

---

> > ### Author Response · Authors · 2025-11-22
> > **Thank you very much for your support and assistance to us! You are our shining light!**
> >
> > # Low-Homophily Justification
> >
> > We provide a **formal bound** (Appendix) showing that the steady-state error of our confidence-gated residual operator satisfies:
> >
> > ∥R⋆ − Y*∥₂ ≤ (∥I − D∥₂ / (1 − ∥D Ã∥₂)) ⋅ ∥R(0) − Y*∥₂
> >
> > This inequality demonstrates that **adaptively reducing confidence cᵢ on nodes with unreliable or heterophilous neighborhoods reduces both the effective propagation norm ∥D Ã∥₂ and the multiplicative error factor**, thereby **limiting error amplification that commonly plagues uniform propagation in low-homophily graphs** (Appendix A.1). We will move this bound and its interpretation into the main text.
> >
> > ---
> >
> > # GAN Stability
> >
> > Our adversarial module uses a **WGAN-GP objective** (Section 3.6) chosen for its **empirical and theoretical stability properties**: the **gradient penalty enforces approximate 1-Lipschitzness of the critic**, which yields **bounded, meaningful gradients to the generator and prevents exploding updates**. We already report **embedding-consistency and perturbation-statistics** showing that **generated perturbations are structured** and that **embedding distortion is reduced (~40.7% improvement vs. ablated model)**. To address the reviewer’s request, we will **add representative optimization traces (critic loss, generator loss, gradient penalty term) and hyperparameter details to the supplementary material**.

---

> > > ### Author Response · Authors · 2025-11-22
> > > **Thank you very much for your support and assistance to us! You are our shining light!**
> > >
> > > Dear Reviewer gH8g, thank you for the careful reading and the constructive comments. Below we summarize our concrete contributions, clarify the evidence supporting our claims, and explain how **PerturbFormer** maps to the **ICLR 2026 subject areas**.
> > >
> > > # Key Contributions and Novel Technical Ideas
> > >
> > > **Propagation-aware adversarial generator (structured topology adversary)**
> > > We propose a **generator conditioned on current node embeddings** that **synthesizes plausible, propagation-relevant topological perturbations** (edge flips or reweightings). This differs from prior work that uses **unconditioned or feature-only adversaries**: our generator explicitly **targets the propagation dynamics** the model uses and thereby creates **meaningful worst-case but realistic corruptions for training**.
> > >
> > > **Confidence-gated residual propagation (per-node adaptive gating)**
> > > We learn **per-node confidence scalars cᵢ** and perform **multi-step residual refinement** via:
> > > **R(t+1) = (I − diag(c))R(0) + diag(c) Ã R(t)**
> > > This **unifies residual diffusion with learned trust**: **unreliable or heterophilous nodes are down-weighted**, reducing **over-smoothing** and **preventing amplification of corrupted signals**.
> > >
> > > **Propagation–adversary co-training (non-additive synergy)**
> > > The **generator and adaptive propagation are co-trained adversarially**: the generator **discovers topology perturbations that exploit current diffusion behavior**, and the propagation **adapts to resist those modes**. **Ablations show the joint training yields larger-than-additive gains**, proving the design is **synergistic**.
> > >
> > > **Heterophily-aware attention and diffusion/attention fusion**
> > > We **inject structural bias into attention logits** and **fuse transformer and diffusion streams**, preserving **expressivity in heterophilous regimes** while maintaining **robustness**.
> > >
> > > **Formal stability analysis**
> > > We provide a **contraction and fixed-point analysis** (Appendix B) showing that, under mild spectral conditions (**∥D Ã∥₂ < 1**), the **adaptive residual operator converges to a unique fixed point** and admits an explicit error bound:
> > > **∥R⋆ − Y*∥₂ ≤ (∥I − D∥₂ / (1 − ∥D Ã∥₂)) ⋅ ∥R(0) − Y*∥₂**
> > > This explains **why per-node gating reduces error amplification on low-homophily graphs**.
> > >
> > > # Fit to ICLR-2026 Subject Areas
> > >
> > > **PerturbFormer** naturally spans multiple **ICLR themes**:
> > >
> > > - **Representation learning and self-supervised learning**: Contrastive pretraining and multi-scale fusion.
> > > - **Generative models**: Propagation-aware generator for structured topology synthesis.
> > > - **Learning on graphs and structured prediction**: Adaptive propagation and heterophily-aware attention.
> > > - **Robustness and uncertainty quantification**: Adversarial topology training and embedding-consistency metrics.
> > > - **Temporal and lifelong learning and optimization**: Contraction conditions for evolving graphs and scalable training recipes.
> > > - **Large-scale learning and infrastructure**: Design choices enabling training on OGB-scale datasets.
> > > # Practical next steps (what we will add in revision)
> > >
> > > Move key spectral/SMB analysis into main text and add an explicit subsection linking heterophily → design → empirical gains.
> > >
> > > Add GAN training curves and GP diagnostics to supplement.
> > >
> > > Include a compact overview diagram in the main paper (from Appendix Fig. A.1) and emphasize Algorithm 1.
> > > # Thank you again for the positive assessment and for pinpointed suggestions that will make the manuscript stronger.

---

> ### Author Response · Authors · 2025-11-27
> **We sincerely hope to receive your support and encouragement!**
>
> # We have addressed the reviewer's concerns and improved our approach. Thank you very much for all the reviewers' suggestions. The latest version has been uploaded and we hope to receive the support and encouragement of all the reviewers, and we sincerely hope your score improvement！

---

### Official Review · Reviewer_7g8J · 2025-10-31

**Soundness:** 1
**Presentation:** 1
**Contribution:** 1
**Rating:** 2
**Confidence:** 3

**Summary:**

The authors summarize six problems in the graph learning domain. To address these issues, they propose PerturbFormer, which integrates multiple techniques (including contrastive learning, attention mechanisms, and adversarial machine learning). In the experiments, the authors report that it demonstrates strong performance.

**Strengths:**

- The field of graph learning is a valuable area.
- The paper utilizes large-scale graph datasets for evaluation.

**Weaknesses:**

- The writing should be substantially improved, as it is very challenging to follow. The subsequent weaknesses stem from this aspect.
- The motivation presented in the paper appears somewhat disorganized. The authors introduce SIX problems in Section 1, but these lack clear organization and detailed explanations.
- The proposed method seems to simply combine existing techniques without introducing significant novelty.
- The writing and clarity of formulas in Section 3 should be enhanced. Additionally, I strongly recommend that the authors provide an overview diagram of the method.

**Questions:**

Please see the weaknesses.

---

> ### Author Response · Authors · 2025-11-22
> **We sincerely hope to receive your support and encouragement!**
>
> We thank the reviewer for the **careful reading**. Below we **respond to the main concerns raised (paraphrased)**, clarifying **misunderstandings** and pointing to **concrete evidence in the manuscript**.
> # After carefully studying the comments, we realized that **many points may include misunderstandings or inaccuracies**, which may have contributed to a **lower evaluation of our submission**.
> # We thank Reviewer 7g8J for the careful reading. We respectfully disagree with several of the reviewer’s broad conclusions and provide precise clarifications, references to analyses already in the submission, and pointers to concrete proofs and ablations below.
> # we sincerely hope your score improvement! We need your support!
> # 1. Reviewer claim ,“The writing should be substantially improved; it is very challenging to follow.”1.
> Response. We appreciate this feedback and will revise wording and layout to improve clarity. That said, the manuscript already contains a clear algorithmic workflow (Algorithm 1) and an overview diagram in the Appendix (Figure A.1 ), which summarize data flow, module interactions and the training loop.
> # 2. Reviewer claim, “The motivation is disorganized; the SIX problems in Sec.1 lack structure and detail.”
>
> Response: We agree that a **tighter presentation of the problem list will improve readability**, and we will **restructure Section 1 in the revision**. To clarify here: the **six challenges** are organized to reflect three categories:
> - **Propagation and adaptation limits** (inflexible propagation, temporal inflexibility)
> - **Structural and statistical brittleness** (robustness deficits, homophily constraints)
> - **Practical constraints** (distillation and condensation limits, computation barriers)
>
> Each category maps directly to a **core design choice in PerturbFormer**:
> - **Confidence-gated residual propagation** addresses propagation and adaptation limits.
> - **Generative adversarially synthesized topology and heterophily-aware attention** address structural brittleness.
> - **Multi-scale condensation and architectural choices** address practical constraints.
>
> See **Introduction and Section 3** for the mapping between **problem categories and design components**.
>
> For quick use in the rebuttal: our motivation is that **realistic robustness requires modeling realistic structural perturbations and coupling that modeling with propagation that is adaptive at the node level**, both are implemented and evaluated in this work (see **Introduction and Section 3** for motivation and method summary).
>
> # 3. Reviewer claim , “The proposed method seems to simply combine existing techniques without introducing significant novelty.”
>
> Response: This is **incorrect**. **PerturbFormer is not a naive composition**. It introduces **three tightly coupled, original contributions**, each **justified, analyzed, and ablated in the paper**:
>
> - **Adaptive Graph-Integrated Label Enhancement**: A **confidence-weighted residual propagation scheme** (see equations in **Section 3 and Appendix**) that learns **per-node confidences cᵢ** and uses them to **gate multi-step residual diffusion**:
>
> **R(t+1) = (I − diag(c))R(0) + diag(c) Ã R(t)**
>
> This operator has an **explicit fixed-point and contraction analysis (Theorem 1)** proving **convergence under mild spectral conditions**. See **Appendix B**.
>
> - **Generative Adversarial Propagation**: A **generator that synthesizes structured, propagation-aware topological perturbations** and a **discriminator that enforces representational stability** to those perturbations. The **generator is conditioned on current embeddings**, producing **realistic perturbations that the propagation mechanism must learn to resist**. This is **different from generic feature-level adversarial regularizers** and is **evaluated against structural-noise baselines**. See **Section 3.6** and the **robustness experiments**.
>
> - **Heterophily-aware attention and ensemble fusion**: A **transformer attention bias** designed to **preserve expressive power in heterophilous neighborhoods** and an **ensemble fusion** that **balances diffusion-based and attention-based predictions** (Eqs. (20)–(21)). This combination **preserves expressivity while controlling over-smoothing**. See **Sections 3.5–3.7**.
>
> These components are **co-trained and interact non-additively**. The **ablation study shows removing GAN or the adaptive residual yields significant drops** (**Full 86.40 → w/o GAN 84.25 / w/o Adaptive propagation 84.91**), proving that the improvements are **not just marginal gains from stacking known blocks**.
> # We sincerely hope to receive your support and encouragement!

---

> > ### Author Response · Authors · 2025-11-22
> > **We sincerely hope to receive your support and encouragement!**
> >
> > # 4.Reviewer claim , “The writing and clarity of formulas in Section 3 should be enhanced. Also strongly recommend an overview diagram.”
> >
> > Response: The **mathematical derivations for the adaptive residual and temporal stability** are provided in **Appendix B (Theorem 1 and the temporal contraction condition)**. We will **bring the key theorem statement and its intuition into the main text** to improve accessibility. The **requested overview diagram already exists in the Appendix (Figure A.1)**, and we will **include a condensed version in the main paper** as requested. See **Algorithm 1 and Appendix B** for the current **formal statements and workflow**.
> >
> > Concretely, **Theorem 1 (Appendix)** states that if **∥Ã∥₂ ≤ 1** and all **cᵢ ∈ (0, 1)**, then the **adaptive residual iteration converges to a unique fixed point**; a **snapshot-wise contraction condition**
> > **maxᵢ cᵢ(τ) · ∥Ã_τ∥₂ < 1**
> > extends this to the **temporal case**. These are **explicit, checkable conditions** (Appendix B).
> > # 5. Reviewer claim, “Novelty / contribution judged poor; soundness poor; experiments unclear.”
> >
> > Response: **Fairness and scope**: All **baselines are trained and evaluated under identical splits and comparable hyperparameter budgets**; **implementation notes and reproducibility details** are provided in the **Appendix**.
> >
> > **Robustness and ablations**: We include
> > - **Component ablation (Table 4)** quantifying the contribution of **GAN** and **adaptive propagation**.
> > - **Systematic structural-noise robustness tests on OGBN-Proteins (Table 5)** showing **substantially smaller AUC degradation than baselines** (maximum −2.05% vs. much larger drops for others).
> > - **Embedding-consistency metrics** showing **GAN reduces embedding distortion by ~40.7% relative to the ablated version**.
> >
> > These results directly support both **soundness** and **empirical novelty**.
> >
> > **Temporal evaluation**: **Incremental learning experiments on TGB–Wikipedia (Table 6)** show **PerturbFormer attains substantially higher final accuracy and far smaller forgetting than temporal baselines**, supporting our **temporal claims**. **Theoretical temporal contraction conditions** are provided in **Appendix B**.
> > # 6. Additional clarifications reviewers sometimes need
> >
> > **Overview figure**: **Appendix A contains Figure A.1 (algorithmic workflow)**; we will **relocate a compact version into the main text** to remove ambiguity.
> >
> > **Convergence and temporal stability**: **Theorem 1 (Appendix B)** and the **snapshot-wise contraction condition** provide the **formal guarantees** and **practical remediation strategies** such as **spectral clipping**, **edge dropout**, and **normalization**.
> >
> > **Ablation and robustness evidence**: **Table 4 (component ablation)**, **Table 5 (structural perturbation robustness)**, and **Table 9 (embedding consistency)** directly **quantify the contributions of the proposed modules**.
> > # 7. Brief motivation paragraph
> > Real graph deployments suffer from **structured topology errors** and **widely varying local reliabilities**. Message passing that treats all nodes uniformly tends to **oversmooth** or **propagate errors**, while **transformers** without **propagation-aware defenses** can be misled by **plausible structural corruptions**. **PerturbFormer** is motivated by the need to **learn realistic structural perturbations** via a **generator conditioned on embeddings**, and **couple those perturbations with a propagation operator** that **adapts per node via learned confidence gating**. This **joint propagation-aware adversarial training** plus **heterophily-aware attention** produces **representations that are robust to topology noise** and **expressive across homophilous and heterophilous regimes**. See **Introduction**, **Section 3**, and **Appendix A** for the workflow and **Section B** for formal guarantees.

---

> ### Author Response · Authors · 2025-11-22
> **We sincerely hope to receive your support and encouragement!**
>
> # 7. Brief motivation paragraph
>
> Real-world graph learning problems are frequently plagued by **structured noise** and **heterogeneous connectivity patterns** that break the assumptions of many established models. **Message-passing and propagation-based methods** excel when **graph topology is reliable and homophily holds**, but they tend to **oversmooth or amplify errors when edges are noisy or misleading**. Conversely, **attention and transformer-style architectures** can **selectively weight neighbors**, yet they are **not explicitly trained to be robust against realistic topological corruptions** and may still be **misled in heterophilous neighborhoods**. Finally, **conventional adversarial defenses** typically **perturb node features or apply generic regularizers**; they **rarely model plausible structural perturbations** that a **propagation operator will actually encounter in deployment**.
>
> **PerturbFormer** is motivated by the observation that **robust graph representation requires modeling realistic structural perturbations and coupling that modeling with propagation that is adaptive to per-node reliability**. Concretely, we view robustness as a **co-training objective**:
>
> - A **generator synthesizes plausible edge perturbations** that reflect the kinds of **topology errors present in target tasks**.
> - A **propagation mechanism learns to resist or correct those perturbations through confidence-gated residual updates**.
>
> Formally, the residual correction is written as:
>
> **R(t+1) = (I − diag(c))R(0) + diag(c) Ã R(t)**
>
> where **cᵢ ∈ (0, 1)** is a **learned per-node confidence** that controls **how much neighborhood information to accept**, and **Ã** denotes the **possibly perturbed normalized adjacency**. Meanwhile, a **learned generator Gϕ produces structured perturbations ΔA conditioned on current embeddings**, creating an **adversarially realistic distribution of topological noise** that the model must withstand.
>
> This joint design **departs from simply stacking existing blocks**:
>
> - The **adversary is propagation-aware** because it **generates perturbations that exploit the current diffusion dynamics**.
> - The **propagation is adversary-trained and node-adaptive** because it **learns to down-weight noisy nodes while preserving reliable signals**.
>
> The net effect is a model that:
>
> - **Learns invariances to plausible structural errors rather than arbitrary noise**.
> - **Reduces harmful oversmoothing by gating propagation at the node level**.
> - **Retains expressivity in heterophilous settings by injecting structural biases into attention**.
>
> These complementary mechanisms **structured perturbation modeling**, **confidence-gated propagation**, and **heterophily-aware attention** are what give **PerturbFormer its empirical robustness and theoretical stability**.
> # 8. On “method is merely a combination of known techniques”
>
> This is incorrect. **PerturbFormer** is more than the sum of its parts. It introduces **three tightly coupled, original mechanisms** and shows they interact **non-additively**.
>
> **Adaptive Graph-Integrated Label Enhancement**. We learn **per-node confidence scalars cᵢ** that explicitly **gate multi-step residual updates**. The update rule is:
>
> **R(t+1) = (I − diag(c))R(0) + diag(c) Ã R(t)**
>
> which **generalizes standard residual propagation**, **enables per-node adaptivity to local reliability**, and **admits a provable fixed point and contraction (Theorem 1)**. The formal proof and discussion appear in **Appendix B**.
>
> **Generative Adversarial Propagation (propagation-aware GAN)**. Unlike **feature-level adversarial regularizers**, our **generator is conditioned on current node embeddings** and **targets topological perturbations that meaningfully challenge the propagation operator**. The **discriminator and adaptive propagation co-train**: the generator **discovers plausible structural corruptions** while the propagation **learns to resist those exact modes of failure**.
>
> **Heterophily-aware attention and ensemble fusion**. We **embed structural bias into attention logits** and **fuse diffusion and attention streams** so the model **preserves expressivity in heterophilous regimes while controlling harmful smoothing**.
>
> Crucially, **ablations show non-additive effects**: removing the GAN or adaptive propagation causes **substantial drops** (**Full 86.40 → w/o GAN 84.25 / w/o Adaptive propagation 84.91 on Proteins**), proving the gains come from the **joint design rather than independent components**.

---

> > ### Author Response · Authors · 2025-11-22
> > **We sincerely hope to receive your support and encouragement!**
> >
> > # 9. On “lack of theory / stability guarantees”
> >
> > We do provide **explicit theoretical guarantees**. **Appendix B** proves that the **adaptive residual operator is a contraction** and **converges to a unique fixed point under mild spectral conditions (Theorem 1)**. For **temporal snapshots**, we propose a **sufficient snapshot-wise contraction condition**:
> >
> > **supₜ maxᵢ cᵢ(τ) ⋅ ∥Ãτ∥₂ < 1**
> >
> > which ensures **per-snapshot contractivity and stable iterative refinement**. **Practical remedies** such as **spectral clipping and normalization** are discussed. These **formal statements and proofs are already included in the submission**.
> > # 10. On experiments / robustness / fairness
> >
> > We ran **extensive evaluations across diverse, large benchmarks** and ensured **fair comparison** with **identical splits** and **comparable hyperparameter budgets** (details in **Appendix**). Key empirical evidence already in the manuscript:
> >
> > **Ablation (component contribution)**: **Table 4** quantifies the contribution of **GAN**, **adaptive propagation**, and other modules.
> >
> > **Robustness under structural perturbations**: **Table 5** reports **relative AUC degradation under deletions, additions, and hybrid noise**. **PerturbFormer degrades far less than baselines** (e.g., **hybrid: −2.05% vs. much larger drops**).
> >
> > **Embedding-consistency metrics**: **GAN reduces embedding distortion by approximately 40.7% compared to the ablated variant**, showing that the **adversarial module preserves representational stability**.
> >
> > **Temporal and incremental learning**: **PerturbFormer attains the highest final accuracy and smallest forgetting on TGB–Wikipedia (Table 6)**
> > # 11. Summary.
> > We respectfully ask the reviewer to reconsider the major negative judgements in light of the **appendix-level overview and algorithm including Algorithm 1 and Figure A.1**, the **formal convergence proofs and temporal contraction condition in Appendix B**, and the **ablation and robustness experiments presented in Tables 4, 5 and 9**.

---

> ### Author Response · Authors · 2025-11-22
> **We sincerely hope to receive your support and encouragement!**
>
> # Why PerturbFormer aligns with ICLR 2026 subject areas
>
> **PerturbFormer** maps directly to multiple **ICLR topics**. Below we list each contribution and the corresponding **ICLR themes**:
>
> **Representation learning and self-supervised learning**: **Contrastive pretraining** and **multi-scale embedding enrichment** improve learned representations (**ICLR: unsupervised / self-supervised representation learning**).
>
> **Generative models**: The **propagation-aware generator** that synthesizes **structural perturbations** is a novel application of **generative modeling to topology robustness** (**ICLR: generative models**).
>
> **Learning on graphs and structured prediction**: The **adaptive residual operator** and **heterophily-aware attention** are fundamental advances in **graph learning and structured prediction** (**ICLR: learning on graphs and other geometries**).
>
> **Robustness, uncertainty, and UQ**: **Explicit adversarial training on plausible topological corruptions** and **embedding-consistency metrics** target **robustness and distributional shifts** (**ICLR: uncertainty quantification, robustness**).
>
> **Large-scale learning and efficiency**: **Mixed-precision training**, **gradient checkpointing**, and **condensed graph-free representations** enable scaling to **OGB and PCQM4Mv2 benchmarks** (**ICLR: large-scale learning, infrastructure**).
>
> **Temporal and lifelong learning**: **Chronological attention kernels** and **snapshot contraction conditions** address **evolving graphs and retention** (**ICLR: lifelong learning, temporal dynamics**).
>
>
> # Core Technical Innovations (What is New)
>
> **Propagation-aware adversarial generator (structured topology adversary)**
> We introduce a **generator conditioned on current node embeddings** that **synthesizes plausible structural perturbations** such as **edge flips and reweightings** that specifically **challenge the propagation operator**.
> **Why novel**: Unlike **generic feature adversaries** or **unconditioned graph corruption**, this generator **models propagation-relevant topology errors**, producing **adversarial examples that the diffusion mechanism must learn to withstand**.
>
> **Confidence-gated residual propagation (per-node adaptive gating)**
> We learn a **per-node confidence scalar cᵢ ∈ (0,1)** and apply it to **gate multi-step residual updates**:
> **R(t+1) = (I − diag(c))R(0) + diag(c) Ã R(t)**
> **Why novel**: This **unifies residual propagation with learned, locality-aware trust**. Nodes **selectively accept neighborhood information**, mitigating **over-smoothing** and **preventing propagation of corrupted signals**.
>
> **Propagation–adversary co-training (non-additive synergy)**
> The **generator and adaptive propagation co-train in an adversarial loop**: the generator **finds topology modes that exploit current diffusion behavior**, and the propagation **adapts to resist those modes**.
> **Why novel**: This creates a **closed feedback loop** where **adversarial examples are propagation-aware**, producing **robustness that cannot be obtained by training components independently**.
>
> **Heterophily-aware attention bias and diffusion/attention fusion**
> We **design attention logits with structural bias tailored for heterophilous neighborhoods** and **fuse diffusion and transformer streams** so the model **retains expressivity without sacrificing robustness**.
> **Why novel**: The design **explicitly balances heterophily expressivity and propagation stability in a unified architecture**.
>
> **Formal stability and convergence analysis for the adaptive operator**
> We provide a **contraction and fixed-point argument** showing the **adaptive residual operator converges under mild spectral conditions** (e.g., **∥Ã∥₂ ≤ 1 and cᵢ ∈ (0,1)**), and extend to a **sufficient snapshot-wise contraction condition for temporal graphs**:
> **supₜ maxᵢ cᵢ(τ) ⋅ ∥Ãτ∥₂ < 1**
> **Why novel**: Theory **links learned per-node gating to provable iterative stability**, a **formal bridge rarely provided in adversarial or robustness work on graphs**.

---

> ### Author Response · Authors · 2025-11-22
> **We sincerely hope to receive your support and encouragement!**
>
> # Empirical and Practical Contributions (What We Deliver)
>
> **Comprehensive robustness evaluation**
> Systematic **structural perturbation tests** (edge deletion, addition, hybrid) show that **PerturbFormer degrades far less than competitive baselines**, demonstrating **practical robustness to topology noise**.
>
> **Ablations validating non-additive gains**
> Removing the **GAN** or **adaptive propagation** yields **substantial performance drops**; results show the **improvements come from the joint design, not marginal stacking**.
>
> **Temporal and incremental learning experiments**
> Empirical tests on **snapshoted and temporal benchmarks** report **improved retention (lower forgetting)** and **higher final accuracy vs. temporal baselines**, corroborating the **temporal stability claim**.
>
> **Scaling and implementation pragmatics**
> Practical design choices such as **multi-hop fusion**, **mixed precision**, **gradient checkpointing**, and **condensed representations** enable **training on large public graph benchmarks** while keeping **runtime and parameter tradeoffs competitive**.
>
> **Representational analyses**
> **Embedding-consistency metrics and visualization** show that the **adversarial module preserves geometry and reduces representational distortion under perturbation**.
>
> **PerturbFormer** makes **three tightly coupled, original contributions**:
>
> ### **Propagation-aware adversarial generator** that **synthesizes realistic topological corruptions conditioned on current embeddings**.
> ### **Learned per-node confidence gating mechanism** for **multi-step residual propagation** with **formal fixed-point and contraction guarantees**.
> ### **Heterophily-aware attention and diffusion fusion** that **preserves expressivity while preventing harmful oversmoothing**.
>
> These components are **co-trained** and exhibit **non-additive improvements** (ablation results confirm **substantial drops when any major module is removed**). Empirically, we demonstrate **robustness to systematic topology perturbations**, **improved temporal retention**, and **competitive scalability on large benchmarks**.
>
> Collectively, these advances map directly to **ICLR 2026 topics**: **representation learning**, **generative models**, **robustness and uncertainty quantification**, **learning on graphs**, and **large-scale learning**. They provide both **theoretical and practical evidence** for the method’s **novelty and impact**.
> # Thank you very much for your support and assistance. We firmly believe that with your suggestions, our paper will be further improved, and we sincerely hope your score improvement.

---

> ### Author Response · Authors · 2025-11-27
> **We sincerely hope to receive your support and encouragement！**
>
> # We have addressed the reviewer's concerns and improved our approach. Thank you very much for all the reviewers' suggestions. The latest version has been uploaded and we hope to receive the support and encouragement of all the reviewers, and we sincerely hope your score improvement！

---

### Official Review · Reviewer_EZoD · 2025-11-01

**Soundness:** 2
**Presentation:** 2
**Contribution:** 3
**Rating:** 4
**Confidence:** 3

**Summary:**

This paper proposes PerturbFormer, a graph representation learning framework that combines many components, such as multi-scale structural embeddings, contrastive pretraining, a heterophily-aware Graphormer and an adversarial propagation mechanism. The core idea is to jointly train a generator that perturbs the graph topology and a discriminator that enforces semantic consistency in node representations. An adaptive residual correction module further refines predictions using node-level confidence scores. The authors evaluate PerturbFormer on a wide range of  datasets and tasks. The results are powerful.

**Strengths:**

1. The paper effectively integrates many components into a single end-to-end framework.
2. This paper is well-organized and easy to follow.
3. The results show strong performance.

**Weaknesses:**

1. The motivation is mentioned briefly but remains underdeveloped. Neither the Abstract nor the Introduction sufficiently clarifies the problem’s significance or the specific gap, and the narrative leans toward describing components.
2. It remains unclear how much each component contributes to the reported gains. The current ablation (Table 4) removes modules singly and therefore does not assess interaction effects.
3. While the method is described in detail, key hyper-parameters are not reported.

**Questions:**

1. In Table 5, PerturbFormer shows remarkable robustness. Is this primarily due to the GAN regularizer, the confidence mechanism, or their combination?
2. The contraction condition (Eq. 26) assumes $\lVert \tilde{A}_{\tau} \rVert_{2} < 1$, but Appendix C shows this often fails in heterophilous graphs. How to address this in practice? Do you apply spectral normalization, and if so, does it degrade performance on homophilous graphs?

---

> ### Author Response · Authors · 2025-11-22
> **We sincerely hope to receive your support and encouragement**
>
> We thank the reviewer for the positive reading and for constructive suggestions that help improve clarity and reproducibility. Below we respond point-by-point to the reviewer’s concerns and questions, and we indicate concrete revisions we will make in the paper.
> We greatly appreciate your support! Therefore, we would prefer to receive further assistance from you.
> # Thank you very much for your support and assistance. We firmly believe that with your suggestions, our paper will be further improved, and we sincerely hope your score improvement.
> # 1. “Motivation is underdeveloped; Abstract/Intro do not sufficiently clarify significance or the specific gap.”
>
> Reviewer comment (paraphrased): The manuscript describes many components but does not clearly articulate why they are necessary together or what exact gap is being closed.
>
> Our response: We respectfully disagree that the motivation is underdeveloped. The submission targets a **concrete, identifiable gap**: existing **propagation and transformer-style models** either **assume homophily and fail to adapt per node** because **residual corrections are global and inflexible**, or **lack explicit mechanisms for representation robustness under structural noise** such as **random or adversarial edge flips**, or **are difficult to scale without sacrificing resilience**. **PerturbFormer addresses all three**. **Node-level confidence calibration** makes **residual correction adaptive to local reliability**. **Adversarial topology synthesis using GAN** explicitly **regularizes the representation to be stable against realistic structural perturbations**. A **heterophily-aware Graphormer** lets **attention incorporate structural bias rather than relying on homophily**. These **design choices and their intended roles** are stated in the **Introduction** and summarized again in the **Method section**. We will **expand the Introduction** in the revision to make the **gap statement** explaining **why per-node adaptation, adversarial robustness, and heterophily bias must be jointly addressed** more **explicit and concise**, and will **add a short paragraph summarizing the single-sentence takeaways for a non-expert reader**.
>
> Gap: Existing **graph-learning approaches** suffer from several persistent limitations: **residual correction schemes that lack node-level adaptivity**, **attention mechanisms that hinge on homophily**, **fragility to structural perturbations**, **challenges in joint graph condensation and distillation**, **scalability bottlenecks on billion-scale graphs**, and **limited adaptability to evolving topologies**.
>
>
> Although prior work has advanced both **propagation-based** and **transformer-based graph learners**, several **critical limitations remain**:
>
> First, **residual correction and smoothing techniques** typically implement **global or uniform propagation rules** and therefore **fail to adapt to node-level uncertainty**, leading to **error amplification where local signals are unreliable**.
>
> Second, many **attention-based transformers** implicitly **rely on homophily assumptions** and **degrade on heterophilous graphs**.
>
> Third, **model robustness to structural perturbations** such as **random or adversarial edge changes** is **rarely addressed systematically**, producing **brittle representations in noisy environments**.
>
> Fourth, **joint condensation or distillation methods** introduce **optimization complexities** that **limit practical utility**.
>
> Fifth, **computational and memory bottlenecks** still **hinder deployment at billion-scale**.
>
> Finally, **existing frameworks often lack temporal flexibility** needed for **evolving topologies**.
>
> These converging gaps motivate an architecture that **calibrates propagation at the node level**, **builds attention mechanisms resilient to heterophily**, **enforces representational stability under structural noise**, and **remains computationally efficient for large and dynamic graphs**, precisely the set of **desiderata we target with PerturbFormer**.

---

> > ### Author Response · Authors · 2025-11-22
> > **We sincerely hope to receive your support and encouragement**
> >
> > # 3. “Key hyper-parameters are not reported.”
> >
> > Reviewer comment (paraphrased): Important hyperparameters such as learning rates, GAN λ, temperature τ, and number of propagation steps T are missing.
> >
> > Our response: We apologize for the omission. **Implementation notes** such as **mixed precision** and **gradient checkpointing** were briefly mentioned, but the **detailed hyperparameter table** was not included in the main paper due to space constraints. We will **add a complete hyperparameter table to the Appendix** in the revision. For transparency, here are the **exact hyperparameters used in the reported experiments** (these settings were used for **OGB and temporal experiments**):
> >
> > - **Optimizer**: AdamW
> > - **Base learning rate (transformer and main networks)**: 1e-3
> > - **Generator/discriminator learning rate**: 5e-4
> > - **Weight decay**: 1e-5
> > - **Batch size (node batches and sampling strategy)**: 1024 effective nodes (sliding window for temporal; neighbor sampling as described)
> > - **Training epochs**: 100 (early stopping on validation), warmup 5 epochs
> > - **Contrastive temperature τ**: 0.1
> > - **Multi-hop K**: K = 3 (concatenate 0..3 hops)
> > - **Residual iterations T**: T = 5 (unless noted otherwise)
> > - **Diffusion γ**: 0.5 (default; varied in sensitivity sweep)
> > - **GAN**: Wasserstein objective, λ (gradient penalty) = 10.0; generator output is edge flip probability Pij (Eq. (15))
> > - **Confidence ceiling**: c̄ = 0.95 (used when enforcing ceilings in high-spectral-radius graphs)
> > - **Mixed precision**: enabled
> > - **Gradient checkpointing**: enabled
> >
> > These **hyperparameters** and a **short paragraph describing tuning ranges and the validation protocol** will be included in **Appendix A (Implementation)** in the revision. We already mention **implementation notes** in the appendix; we will **expand that section with a formal table**.
> > # 4. “In Table 5 PerturbFormer shows remarkable robustness. Is this primarily due to the GAN regularizer, the confidence mechanism, or their combination?”
> >
> > Reviewer question (paraphrased): Which module is the dominant source of robustness?
> >
> > Our response: **Robustness in our framework results from both components performing distinct but complementary roles**.
> >
> > **GAN regularizer (representation stabilization)**: By synthesizing plausible perturbations and training the model to be invariant to them, the GAN **reduces embedding distortion under structural noise** as shown in **Table 9 (Rabs and Rrel improvements)**, which means it improves **representational stability**.
> >
> > **Confidence-weighted residual correction (prediction refinement)**: The residual propagation **refines label estimates with node-level confidence cᵢ**, preventing **error amplification from uncertain neighbors**. The **theoretical fixed-point result (Appendix B)** shows the residual iteration **converges under the contraction condition (maxᵢ cᵢ · ||Ã||₂ < 1)**, which means **confidence scalars control propagation strength and stability**.
> >
> > **Empirical evidence**:
> > - **Table 4** shows removing GAN drops accuracy by **2.15% (86.40 → 84.25)** and removing adaptive propagation drops it by **1.49% (86.40 → 84.91)**. Each alone causes a **sizable degradation**, indicating **both contribute significantly**.
> > - **Table 5 (relative ROC–AUC degradation under perturbations)** shows **PerturbFormer’s degradation is much smaller than baselines (maximum ≈ −2.05%)**, which is consistent with the GAN improving **representation invariance** and the confidence propagation **preventing incorrect diffusion of noisy signals**.
> >
> > **Conclusion**: The robustness reported in **Table 5** is **not attributable to a single silver-bullet module**. The **GAN and confidence-weighted correction play distinct but synergistic roles**. We will **add a short experiment table** in the revision that **isolates the representation stability impact (embed-shift metrics) versus prediction refinement impact (accuracy under perturbation)** to make this **complementarity explicit**.

---

> > > ### Author Response · Authors · 2025-11-22
> > > **We sincerely hope to receive your support and encouragement**
> > >
> > > # 5. “The contraction condition (Eq. 26) assumes ||Ã_τ||_2 < 1, but Appendix C shows this often fails in heterophilous graphs. How to address this in practice? Do you apply spectral normalization, and if so, does it degrade performance on homophilous graphs?”
> > >
> > > Reviewer comment (paraphrased): The theoretical contraction condition sup_τ maxᵢ cᵢ^{(τ)} · ||Ã_τ||₂ < 1 may not hold in heterophilous graphs. How is this handled practically, and what are the tradeoffs?
> > >
> > > Our response: **Clarification**: We presented the contraction condition as a **sufficient criterion for stability** in **Appendix B (Theorem and temporal condition)**. The appendix explicitly acknowledges that **real heterophilous graphs can violate ||Ã||₂ ≤ 1** and provides **practical remedies**. See **Appendix B** and the section **“Practical remarks on spectral bounds.”**
> > >
> > > **Practical measures we use and recommend**:
> > > - **Empirical spectral check and light scaling**: Before training, we compute an **empirical bound on ||Ã||₂**. If it exceeds 1 by a meaningful margin, we **scale the normalized adjacency by a factor s ≤ 1 (light spectral clipping)** so that **||s·Ã||₂ ≲ 1**. This preserves **relative connectivity** while satisfying the sufficient condition.
> > > - **Small confidence ceilings**: We clip learned confidence scores with **cᵢ ← min(cᵢ, c̄)** where **c̄ < 1** (we used **c̄ = 0.95** in some high-spectral-radius graphs). This reduces **effective propagation strength** and preserves **contractivity in practice**.
> > > - **Edge dropout and spectral regularizers**: During training, we sometimes use **light edge dropout** or add a **small spectral regularizer** to discourage excessively large singular values.
> > >
> > > **Does spectral scaling hurt homophilous graphs?** In our empirical checks, **light spectral scaling (factor s close to 1)** had **negligible effect on performance in homophilous graphs** while enabling **stable iterations in heterophilous settings**. We therefore **do not apply heavy spectral normalization by default** and only use the **minimal transformation necessary when the empirical spectral radius threatens contractivity**.
> > >
> > > The **Appendix explicitly suggests these practical remedies** (spectral clipping, confidence ceilings). We will **add an ablation in the revision** that measures **performance with and without light scaling on both homophilous and heterophilous benchmarks** to quantify the tradeoff.
> > >
> > > # 6. Minor clarifications & clarifying statements we will incorporate
> > >
> > > We will **expand the Introduction** to state the **gap more directly** and provide a **one-paragraph summary explaining why these components work together** to address the reviewer’s point.
> > >
> > > We will **add a detailed hyperparameter table** and a **short paragraph on tuning protocol** to **Appendix A (Implementation notes)**.
> > >
> > > We will **include explicit pairwise ablation results** (GAN and adaptive propagation combinations) as a **table in the revision** so that reviewers and readers can **inspect interaction effects directly**. The current paper already provides **quantitative evidence (Table 4)** and **embedding-stability evidence (Table 9)** showing that the modules are **complementary**; we will make the **interaction analysis explicit**.
> > >
> > > We will **add a short appendix subsection** showing the **empirical spectral radii measured on our benchmarks** and giving the **exact light-scaling multipliers used in experiments where scaling was necessary**. The **Appendix already discusses spectral bounds and heterophily statistics**, and we will make this **more concrete**.
> > > # 7. Short technical pointers / reminders for the reviewer (where to find material in our submission)
> > >
> > > **Component ablation (Table 4)**: Proteins test accuracies and per-module ablation numbers are reported in the **main paper (Section 4.3, Table 4)**.
> > >
> > > **Robustness under structural perturbations (Table 5)**: Relative **ROC–AUC degradation numbers** are provided in **Section 4.4, Table 5**.
> > >
> > > **Embedding stability metrics (Table 9)**: **Absolute and relative embedding shifts (Rabs, Rrel)** supporting the **GAN’s representational effect** are shown in the **Appendix**, along with **comparative robustness visualization**.
> > >
> > > **Contraction theorem and temporal contraction condition (Eq. 26, Appendix B)**: The **fixed-point theorem** and the **snapshot-wise contraction condition (maxᵢ cᵢ^{(τ)} · ||Ã_τ||₂ < 1)** together with **practical remarks** appear in **Appendix B**. We explicitly discuss **how to enforce or approximate this condition in practice**.

---

> > ### Author Response · Authors · 2025-11-22
> > **We sincerely hope to receive your support and encouragement**
> >
> > # 2. “It’s unclear how much each component contributes, Table 4 only removes modules singly and doesn’t assess interaction effects.”
> >
> > Reviewer comment (paraphrased): The current ablation only turns off modules one at a time; interactions among modules are not inspected.
> >
> > Our response: We agree that **interaction effects are an important diagnostic**. The **single-module ablations in Table 4** were intended to show that **each component is necessary**. The manuscript also includes **complementary embedding-stability metrics** in **Table 9 (embedding consistency)** which provide **orthogonal evidence about how the GAN affects learned geometry** by showing **reduced embedding shift**.
> >
> > **Table 4 (Proteins)** shows clear **per-module drops** when removing **GAN regularization (86.40 → 84.25)** or **adaptive propagation (86.40 → 84.91)**, demonstrating that **both modules are individually important**. **Table 9 (embedding consistency under δ = 10% hybrid perturbation)** shows that **adding GAN reduces absolute and relative embedding distortion (Rabs 15.02 → 8.91, Rrel 0.21 → 0.13)**, supporting the claim that **the GAN stabilizes representations so downstream residual correction works on more stable embeddings**.
> >
> > These two lines of evidence together indicate **complementarity**: **the GAN reduces embedding distortion (representation-level robustness)** while **adaptive residuals refine noisy label signals at the prediction level**. That combination produces **greater end-to-end robustness than either alone**, which is implicitly supported by the above tables. We will **add an explicit pairwise ablation table** in the revision (**GAN off + adaptive propagation off together, and GAN off + attention bias off, etc.**). However, the current material already provides **both accuracy and embedding-stability evidence** that the two modules act in **complementary ways**.

---

> ### Author Response · Authors · 2025-11-22
> **We sincerely hope to receive your support and encouragement**
>
> # 8. Summary
> We appreciate the reviewer’s careful reading. The questions raised improved our awareness of where **clarifications and additional empirical diagnostics** are needed. Concretely, we will **strengthen the motivation in the Introduction**, **add a hyperparameter table and tuning details to Appendix A**, **include explicit pairwise ablation numbers in a new table**, and **add an empirical section documenting spectral radii and the light-scaling factors used**, along with **ablations to show light scaling’s negligible effect on homophilous graphs**. The manuscript already contains **proofs for residual convergence** and a **practical discussion of spectral remedies**; we will **expand those and move selected items into the main text where space permits**.
>
> # Thank you for the careful review and constructive suggestions. We believe the clarifications and targeted new material described above directly address the reviewer’s concerns and will make the manuscript stronger and easier to evaluate.

---

> ### Author Response · Authors · 2025-11-22
> **Novelty, Problems Addressed, Contributions, and ICLR Fit**
>
> **PerturbFormer** introduces a **unified framework** that improves both **accuracy** and **robustness** of **graph representation learning** by:
> ### **Jointly training an adversarial topology generator and a discriminator** to synthesize and regularize realistic structural perturbations.
> ### **Integrating a heterophily-aware Graphormer** to make attention structure-sensitive.
> ### **Learning multi-scale structural embeddings via contrastive pretraining** to stabilize geometry.
> ### **Applying a node-confidence-weighted residual correction mechanism** with **provable convergence properties** and **practical remedies**.
>
> Together these elements address three concrete gaps:
> - **Lack of per-node adaptive propagation**
> - **Brittleness of transformer-style attention under heterophily and structural noise**
> - **Absence of end-to-end adversarial regularization for topology**
>
> This design produces **consistent improvements in robustness and accuracy across diverse benchmarks**.
>
> ---
>
> # Key Innovations (What’s New)
> - **End-to-end adversarial topology regularization**: Unlike static augmentations or post-hoc robustness tests, we train a topology generator and a discriminator jointly with the representation learner so the model learns invariance to plausible, model-informed structural perturbations.
> - **Heterophily-aware Graphormer**: We adapt transformer-style attention with structure-sensitive biases so that attention mechanisms do not implicitly assume homophily, improving performance on non-homophilous graphs.
> - **Multi-scale contrastive structural pretraining**: Combining multi-hop structural encodings with contrastive objectives produces embeddings that are empirically more stable under topology changes.
> - **Node-confidence-weighted residual correction with theoretical guarantees**: We propose a per-node confidence mechanism to adaptively control propagation strength and provide a convergence analysis together with practical mitigation strategies (spectral clipping, confidence capping, light edge dropout).
>
> ---
>
> # Problems Addressed (Why These Innovations Matter)
> - **Uniform propagation amplifies errors**: Existing residual or diffusion schemes typically use uniform or global rules and therefore risk spreading errors from low-confidence nodes. Our per-node confidence gating prevents error amplification.
> - **Transformer brittleness in heterophilous graphs**: Standard attention often underperforms when node labels are not correlated with local connectivity. Structure-aware attention remedies this mismatch.
> - **Lack of training-time structural robustness**: Most approaches evaluate robustness only at test time. Our generator–discriminator training embeds robustness into representations during learning.
> - **Theory–practice gap for iterative propagation**: We supply formal contraction conditions and practical workarounds so the iterative residual correction is stable and applicable to real-world heterophilous graphs.
>
> ---
>
> #  Main Contributions (Concrete Deliverables)
> - **PerturbFormer**, a modular end-to-end architecture that couples adversarial topology synthesis, heterophily-aware attention, contrastive multi-scale pretraining, and adaptive residual correction.
> - **Theoretical analysis** showing sufficient conditions for convergence of the confidence-weighted residual iteration and pragmatic strategies when spectral bounds are violated.
> - **Extensive empirical evaluation** demonstrating consistent accuracy and robustness gains over baselines on homophilous, heterophilous, and temporally evolving benchmarks, including embedding-stability metrics.
> - **Ablation and diagnostics** that identify the distinct roles of GAN-based regularization (representation stability) and confidence-weighted propagation (prediction refinement); we will extend these with pairwise interaction ablations in revision.
> - **Reproducibility artifacts** (implementation notes, hyperparameter settings, and practical recipes) to enable community follow-up.
>
> ---
>
> #  Why PerturbFormer Fits ICLR
> PerturbFormer intersects multiple central **ICLR themes**:
> - **Representation learning** (contrastive pretraining and embedding stability)
> - **Generative models** (adversarial topology synthesis)
> - **Learning on graphs and geometry**
> - **Robustness and uncertainty quantification** (confidence gating and adversarial training)
> - **Scalability** (design choices and practical tricks targeting large graphs)
>
> It therefore offers **methodological, theoretical, and empirical insights** of clear interest to the ICLR community.
> #  Thank you very much for your support and assistance. We firmly believe that with your suggestions, our paper will be further improved, and we sincerely hope your score improvement.

---

> ### Author Response · Authors · 2025-11-27
> **We sincerely hope to receive your support and encouragement**
>
> # We have addressed the reviewer's concerns and improved our approach. Thank you very much for all the reviewers' suggestions. The latest version has been uploaded and we hope to receive the support and encouragement of all the reviewers, and we sincerely hope your score improvement！

---

### Note · Authors · 2026-01-27

**Comment:**

I have read and agree with the venue's withdrawal policy on behalf of myself and my co-authors.

**Withdrawal Confirmation:**

I have read and agree with the venue's withdrawal policy on behalf of myself and my co-authors.

---

### Meta-Review · Area_Chair_91oK · 2026-01-01

**Summary:**

In this paper, the authors propose PerturbFormer, a graph representation learning framework, aiming to address the transformer attention degradation under low homophily, vulnerability to structural perturbations, and the high cost of large-scale inference. Experimental results are provided to support the proposed method.

The reviewers generally agree that the paper presents a technically reasonable solution. However, they also raised several critical concerns, including the technical novelty of the proposed method, the reproducibility of the experimental results, and, most importantly, the presentation issues. The authors provided a rebuttal, but the reviewers did not (get to) respond. I briefly checked the paper and found the issues indeed exist: at least the presentation of the paper is indeed confusing, and many descriptions are hard to follow. Considering the rebuttal and the discussions (including those that could have taken place), the paper is currently clearly below the bar for acceptance, and I vote for rejection.

Besides, the ICLR Code of Conduct indicates that the authors should not exploit the identity of reviewers, and the authors provide some  highly unprofessional comments, which will be reported to SAC/PCs.

**Reviewer Concerns:**

Reviewer EZoD:
W1: The motivation is mentioned briefly but remains underdeveloped.
The authors responded, but the concern remains valid.

W2: It remains unclear how much each component contributes to the reported gains.
The authors responded, but the concern remains valid.

W3: Key hyper-parameters are not reported.
The issue is partially addressed.


Reviewer 7g8J:
W1: The writing should be substantially improved.
The authors responded, but the concern remains valid.

W2: The motivation presented in the paper appears somewhat disorganized.
The authors responded, but the concern remains valid.

W3: The proposed method does not have significant novelty.
The authors responded, but the concern remains valid.

W4: The writing and clarity of formulas in Section 3 should be enhanced.
The authors responded, but the concern remains valid.

Reviewer gH8g:
W1: Claims of outperforming prior work in low-homophily settings are not rigorously.
The issue is partially addressed.

W2: There is no experimental curves or metrics on adversarial optimization progress.
The issue is partially addressed.

Reviewer 4Gtn:
W1: The first major weakness of this paper lies in its writing quality.
The authors responded, but the concern remains valid.

W2: The approach PerturbFormer lacks novelty.
The authors responded, but I doubt the reviewer will be convinced.

W3: The experimental setup is confusing.
The authors responded, which partially addressed the issue.

W4: The reported results are hard to believe.
The authors responded, but the concern remains valid.

W5: No analysis of its computational complexity and runtime comparison.
The authors responded, which partially addressed the issue.

**Reviewer Scores:**

For Reviewer EZoD, the initial rating is 4, and it is likely to stay at 4.

For Reviewer 7g8J, the initial rating is 2, and it is likely to stay at 2.

For Reviewer gH8g, the initial rating is 8, and it is likely to stay at 8.

For Reviewer 4Gtn, the initial rating is 2, and it is likely to stay at 2.

---

### Decision · Program_Chairs · 2026-01-26

Reject